

# Engineering unique localization transition with coupled Hatano-Nelson chains

Ritaban Samanta[*], Aditi Chakrabarty[†] and Sanjoy Datta[‡]

Department of Physics and Astronomy, National Institute of Technology,
Rourkela, Odisha-769008, India

[*] ritabansamanta28@gmail.com , [†] aditichakrabarty030@gmail.com , [‡] dattas@nitrkl.ac.in

## Abstract

The paradigmatic Hatano-Nelson (HN) Hamiltonian induces a delocalization-localization (DL) transition in a one-dimensional (1D) lattice with random disorder, in striking contrast to its Hermitian counterpart. The DL transition also persists in the presence of a quasiperiodic potential separating completely delocalized and localized eigenstates. In this study, we reveal that coupling two 1D quasiperiodic Hatano-Nelson (QHN) lattices significantly alters the nature of the DL transition and identify two critical points, $V_{c1} < V_{c2}$, when the nearest neighbors of the two 1D QHN lattices are cross-coupled with strong hopping amplitudes under periodic boundary conditions (PBC). Complete delocalization occurs below $V_{c1}$ and the states are completely localized above $V_{c2}$, while two mobility edges symmetrically emerge about $\text{Re}[E] = 0$ between $V_{c1}$ and $V_{c2}$. Notably, under specific asymmetric cross-hopping amplitudes, $V_{c1}$ approaches zero, resulting in localized states even for an infinitesimally weak potential. Remarkably, we also find that the mobility edges precisely divide the delocalized and localized states in equal proportions. We demonstrate a possible implementation of these findings in a coupled waveguided array which can be exploited to control and manipulate the light localization depending upon the hopping amplitude in the two QHN chains.

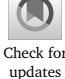

# 1  Introduction

The concept of localization of the matter waves was laid down by P.W. Anderson in 1958, wherein the investigation revealed that in the presence of a sufficiently strong random disorder in the 3D lattice, the electronic conductivity ceases, hence becoming an insulator (frequently termed as the *Anderson* localization) [1]. The interesting features of Anderson localization has been implemented in many domains of physics, such as superconductors [2–4], photonics [5–8] and acoustics [9, 10]. However, it was later demonstrated using a scaling law that in the lattices of lower dimensions (1D/2D), even an infinitesimally small strength of the random disorder localizes all the electronic wave functions [11]. A few years later, in 1980, S. Aubry and G. André demonstrated that in quasiperiodic lattices, a delocalization-localization (DL) transition takes place even in lower dimensions [12,13]. In the cosine-modulated Aubry-André-Harper (AAH) models, the DL transition occurs at a finite value of the quasiperiodic potential, governed by the self-duality of the Hamiltonian in the real and momentum spaces [14]. For closed quantum systems which are described by Hermitian Hamiltonians, there have been many works based on the AAH model in the last few years [15–24]. Recently, such quasiperiodic lattices have been realized in the ultracold atomic systems [25–27].

However, in reality, most of the condensed matter systems are coupled to the environment that exchanges either energy, or particles, or both with the surroundings. Such open systems are frequently mapped using a non-Hermitian Hamiltonian. Hatano and Nelson in 1996 introduced one such model which is an extension of the Anderson model with asymmetric hopping amplitudes. In his work, originally on the superconductors, it was shown that in the presence of such random disorder, the DL transition is manifested in 1D systems. There have been many ongoing studies on the localization, spectral properties, self-duality and mobility edges in various non-Hermitian systems [17, 28–31]. Besides, such systems with asymmetric hopping amplitudes have been gaining attention over the years due to the phenomenon of skin effect wherein a macroscopic number of bulk states become localized at one of the edges under open boundaries [32–35].

On the other hand, some recent works have been carried out on coupled AAH chains in which two disparate chains of atoms are coupled to each other by some interchain hopping amplitudes [36, 37]. It was demonstrated that such a coupled Hermitian AAH chain shows interesting properties like the existence of mobility edges. However, to the best of our knowledge, the interplay of the quasiperiodicity and the coupling between the non-Hermitian chains of Hatano-Nelson(HN) type have not been investigated so far. Therefore, the aim of this work is to investigate a coupled HN bipartite chain in the presence of AAH type potential to closely scrutinize the localization behavior in such coupled systems and engineer unique localization features owing to the combined effect of the strength of the intra/inter chain coupling and the quasiperiodic potential. Intriguingly, we find that the presence of a strong interchain coupling between two dissimilar atoms in the two sublattices possessing symmetric and asymmetric in-

terchain hopping between two atoms of adjacent unit cells render equal proportion of localized and delocalized states in the presence of quasiperiodic potential. Moreover, we find that in the latter case, half of the states are localized even for a very weak strength of the quasiperiodic potential, akin to the study by Anderson on 1D systems. Similar to the previous investigations, we illustrate that the completely delocalized and localized eigenstates possess entirely complex and real spectra respectively [38, 39], whereas the intermediate regime demonstrates a mixture of real and complex eigenenergies due to the presence of both delocalized and localized eigenstates. We reveal that the coupling renders distinct properties in the skin effect as compared to the conventional HN systems, wherein some of the localized states (in the bulk) under PBC become skin-states under the OBC. Finally, we suggest a possible experimental set-up in coupled optical waveguides to exploit the uniqueness of the localization of eigenstates at arbitarily small strength in the quasiperiodic potential.

This work is organized as follows: In Sec.2.1, we discuss the coupled QHN Hamiltonian and elaborate the method to numerically identify the delocalized and localized phases in Sec. 2.2. We analytically determine the strength of the quasiperiodic potential ($V_{c1}$ and $V_{c2}$) where the localization transitions occur in Sec. 3. In Sec. 4, we demonstrate our unique findings in the presence of various ratios of the strong interchain coupling between the two QHN chains. We propose a feasible experimental set-up in coupled optical waveguides in Sec. 5. Finally, Sec. 6 consists of a summary of the work, highlighting the important results and unique findings.

## 2 Model and methods

### 2.1 The coupled QHN Hamiltonian

We consider two uni-directional HN chains with quasiperiodic potential (consisting of two sublattices A and B in a single unit cell) coupled to each other *via*. an interchain hopping, which we call a coupled quasiperiodic HN (QHN) Hamiltonian from here on.

The Hamiltonian in such a coupled system is given by,

$$\mathcal{H} = \mathcal{H}_A + \mathcal{H}_B + \mathcal{H}_C \,, \tag{1}$$

where, $\mathcal{H}_A(\mathcal{H}_B)$ is the Hamiltonian for chain 1(2) of atom A(B) and $\mathcal{H}_C$ introduces the interchain coupling between chains 1 and 2. The individual terms of the Hamiltonian are described as,

$$\mathcal{H}_{A(B)} = \sum_{n=1}^{N-1} \left( t_R c_{n+1,A(B)}^\dagger c_{n,A(B)} + t_L c_{n,A(B)}^\dagger c_{n+1,A(B)} \right) + \sum_{n=1}^{N} V \cos(2\pi n\alpha) c_{n,A(B)}^\dagger c_{n,A(B)} \,. \tag{2}$$

Figure 1: Schematic diagram of the coupled QHN model. Atoms *A* are depicted in green, and atoms *B* are depicted in purple. The n*th* unit cell containing the two atoms is demonstrated by the dashed rectangle. The different interchain hopping amplitudes are mentioned below and are represented by coloured arrow lines.

Here, $c^\dagger_{n,x}(c_{n,x})$ are the fermionic creation (annihilation) operators at the site $n$ of sublattice $x = A(B)$. The first two terms of the Hamiltonian $H_{A(B)}$ define the usual asymmetric intrachain hopping of the fermions between the nearest neighbour sites in sublattices $A(B)$ and the second term is the onsite quasiperiodic potential. $\alpha$ is an irrational number approximated as $F_{n-1}/F_n$, where $F_n$ and $F_{n-1}$ are the $n$th and (n-1)$th$ terms of the Fibonacci series respectively. Throughout this work, we have considered $\alpha$ to be $(\sqrt{5}-1)/2$ which approximates the inverse golden mean ratio. The final part of the Hamiltonian which couples the two distinct HN chains $via$. interchain coupling amplitudes is given as,

$$\mathcal{H}_C = \sum_{n=1}^{N}\left(u_R c^\dagger_{n+1,A}c_{n,B} + u_L c^\dagger_{n,B}c_{n+1,A} + w_R c^\dagger_{n+1,B}c_{n,A} + w_L c^\dagger_{n,A}c_{n+1,B}\right). \tag{3}$$

The interchain coupling $u_R(u_L)$ is the hopping strength from $B_n \rightarrow A_{n+1}(A_{n+1} \rightarrow B_n)$, whereas $w_R(w_L)$ is the hopping strength of $A_n \rightarrow B_{n+1}(B_{n+1} \rightarrow A_n)$. All these terms of the inter and intra chain coupling are depicted in a schematic in Fig. 1.

## 2.2 Delocalization-localization (DL) transition: The IPR

The localized and delocalized behaviour of the eigenstates of the system is characterised by estimating the value of the Inverse Participation Ratio (IPR). The IPR for a given eigenstate ($m$) is given by [40],

$$IPR_m = \frac{\sum_{n=1}^{N}\sum_{x=A,B}|\psi^m{}_{n,x}|^4}{(\sum_{n=1}^{N}\sum_{x=A,B}|\psi^m{}_{n,x}|^2)^2}, \tag{4}$$

where, $\psi^m{}_{n,x}$ is the normalized wave function of eigenstate labelled by $m$ at site $n$ for the chain $x = A, B$. Here, $N$ is the size of the system and the number of total eigenstates is given by $L = 2N$. It is well known that for the delocalized states, the $IPR$ varies as $IPR \sim L^{-1}$. In the thermodynamic limit ($N \rightarrow \infty$), and therefore $IPR \sim 0$. In contrast, for the localized states, the $IPR$ is independent of the system size and approaches 1 in the thermodynamic limit. All our numerical estimates are for a lattice with 610 sites, unless specifically mentioned.

# 3 Analytical understanding of the localization transition

In the following discussion, we analytically estimate the critical value of quasiperiodic potential for the DL transition in the coupled QHN Hamiltonian as defined in Sec. 2.1. The Hamiltonian consists of creation and annihilation operators of two sublattices, which can be effectively combined into a single equation in terms of a spinor representation [41] given as,

$$b = \begin{pmatrix} c_A \\ c_B \end{pmatrix}. \tag{5}$$

Using Eq. (5), one can immediately obtain,

$$\mathcal{H} = \sum_{n=1}^{N}\left(b^\dagger_n T_1 b_{n+1} + b^\dagger_{n+1} T_2 b_n\right) + \sum_{n=1}^{N} b^\dagger_n \epsilon(n) b_n. \tag{6}$$

Here,

$$\epsilon(n) = \begin{pmatrix} V_n & 0 \\ 0 & V_n \end{pmatrix}, \tag{7}$$

where $V_n$ is the onsite quasiperiodic potential at the $n^{th}$ lattice site defined as $V\cos(2\pi n\alpha)$. We now introduce the transfer matrices under PBC as,

$$T_1 = \begin{pmatrix} t_L & w_L \\ u_L & t_L \end{pmatrix}, \qquad T_2 = \begin{pmatrix} t_R & u_R \\ w_R & t_R \end{pmatrix}. \tag{8}$$

We introduce the wave function as,

$$\psi_n^m = \begin{pmatrix} \psi_{n,A}^m \\ \psi_{n,B}^m \end{pmatrix}, \tag{9}$$

where, $\psi_{n,x}^m$ is the normalized wave function of eigenstate labelled by $m$ at site $n$ for the chain $x = A, B$. Substituting Eq. (9) in Eq. (6), we obtain,

$$\left(E_m \mathbb{1} - \epsilon(n)\right)\psi_n^m = T_1 \psi_{n+1}^m + T_2 \psi_{n-1}^m. \tag{10}$$

Eq. (10) can be disintegrated into the following coupled equations:

$$\left(E_m - V_n\right)\psi_{n,A}^m = t_L \psi_{n+1,A}^m + t_R \psi_{n-1,A}^m + w_L \psi_{n+1,B}^m + u_R \psi_{n-1,B}^m, \tag{11}$$

and

$$\left(E_m - V_n\right)\psi_{n,B}^m = t_L \psi_{n+1,B}^m + t_R \psi_{n-1,B}^m + w_R \psi_{n-1,A}^m + u_L \psi_{n+1,A}^m. \tag{12}$$

Applying the following canonical transformation

$$\psi_n^{m\pm} = \frac{\psi_{n,A}^m \pm \psi_{n,B}^m}{\sqrt{2}}, \tag{13}$$

and with the following restrictions, i.e., $u_R = w_R = u_1$ and $u_L = w_L = u_2$, the system can be exactly mapped to two uncoupled QHN chains. This can be explicitly written as,

$$\left(E_m - V_n\right)\psi_n^{m+} = \left(t_L + u_2\right)\psi_{n+1}^{m+} + \left(t_R + u_1\right)\psi_{n-1}^{m+}, \tag{14}$$

and

$$\left(E_m - V_n\right)\psi_n^{m-} = \left(t_L - u_2\right)\psi_{n+1}^{m-} + \left(t_R - u_1\right)\psi_{n-1}^{m-}. \tag{15}$$

The full spectrum is therefore composed of the spectra of the two uncoupled QHN chains, i.e., $E_m^- = E_m - V_n$ and $E_m^+ = E_m - V_n$, which are identical. From Eqs. 14 and 15, one expects two localization transitions at two critical strengths of the quasiperiodic potential at [42],

$$V_{c1} = 2\Big[\max(|t_L - u_2|, |t_R - u_1|)\Big], \tag{16}$$

and

$$V_{c2} = 2\Big[\max(|t_L + u_2|, |t_R + u_1|)\Big]. \tag{17}$$

$V_{c1}$ provides the maximum value of quasiperiodic potential below which all the eigenstates are delocalized. $V_{c2}$ is that strength of the potential above which all the states become completely localized. It is interesting to note that one can engineer a system where $V_{c1}$ is zero, when the conditions $|t_L - u_2| = 0$ and $|t_R - u_1| = 0$ are simultaneously satisfied.

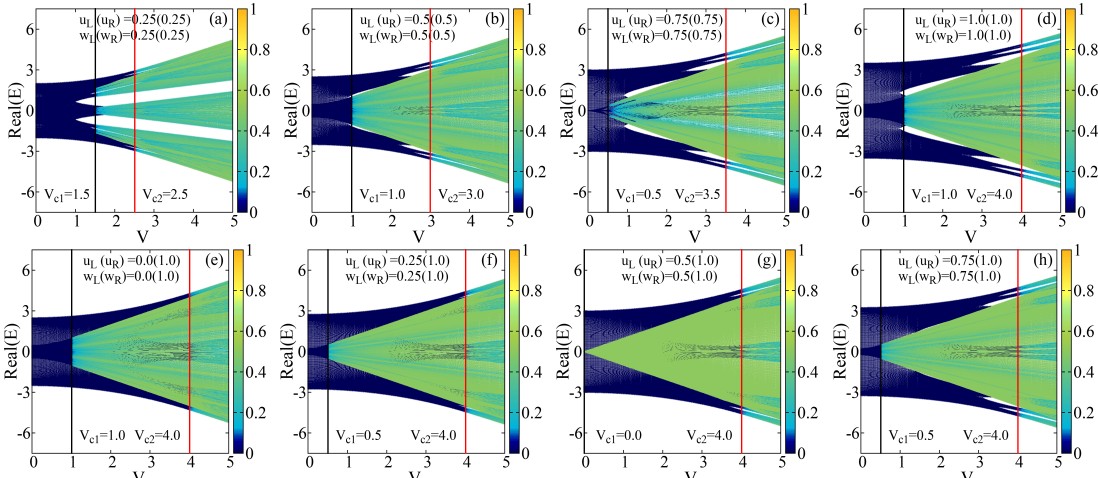

Figure 2: The localization behavior in a strongly coupled QHN Hamiltonian with $t_L = 0.5, t_R = 1.0$ in a lattice with $N = 610$ sites under PBC with interchain hopping between the two adjacent unit cells. Projection of IPR as a function of the real part of the eigen-energy and quasiperiodic potential ($V$) for (a-d) symmetric interchain hopping and (e-h) asymmetric interchain hopping. In both the panels, the DL transition is shown as a dark blue to green transition. In particular, the parameters of interchain coupling are: (a) $u_L = u_R = w_L = w_R = 0.25$, (b) $u_L = u_R = w_L = w_R = 0.5$, (c) $u_L = u_R = w_L = w_R = 0.75$, (d) $u_L = u_R = w_L = w_R = 1.0$, (e) $u_L = w_L = 0.0$ and $u_R = w_R = 1.0$, (f) $u_L = w_L = 0.25$ and $u_R = w_R = 1.0$, (g) $u_L = w_L = 0.5$ and $u_R = w_R = 1.0$ (h) $u_L = w_L = 0.75$ and $u_R = w_R = 1.0$.

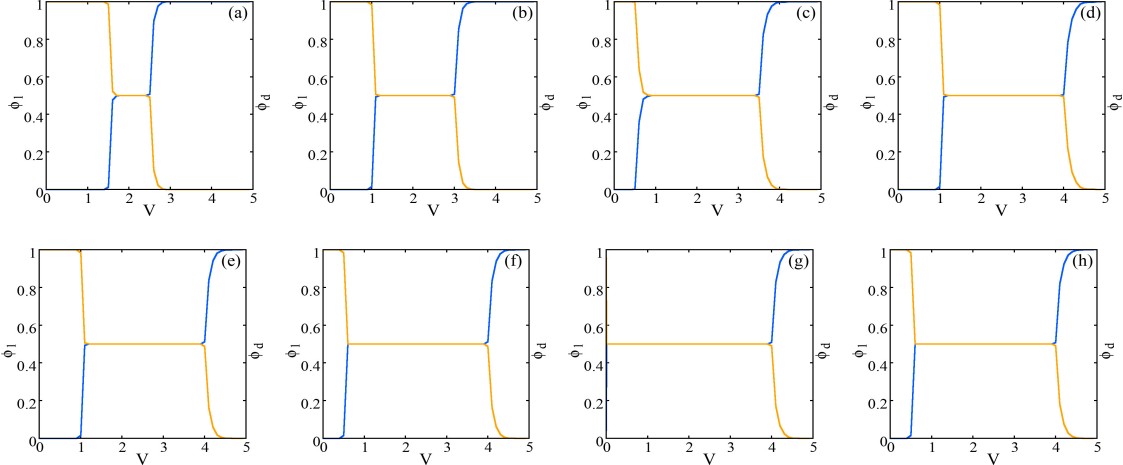

Figure 3: The fraction of localized states ($\phi_l$) in blue and delocalized states ($\phi_d$) in yellow corresponding to the parameters of Fig 2. States with $IPR \gtrsim 0.1$ was considered localized, otherwise the states are considered to be delocalized in nature. We have used PBC and the remaining parameters are same as in Fig. 2.

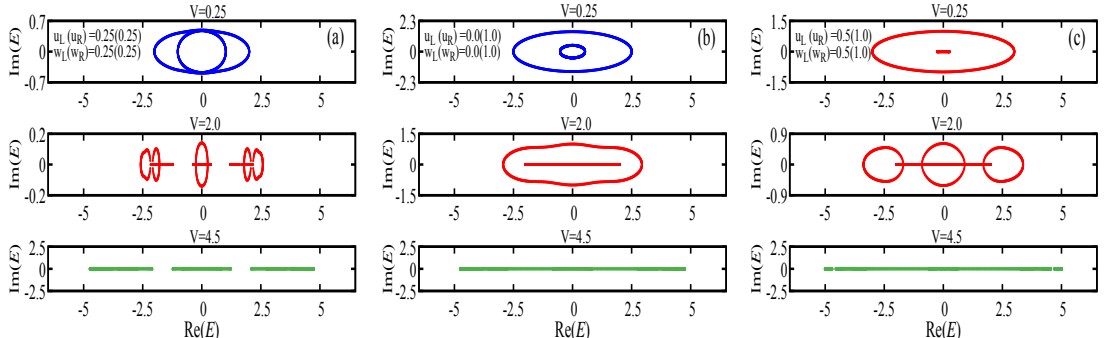

Figure 4: (a), (b), (c) The energy spectrum in the complex plane under the PBC in the three regimes of the phase diagram corresponding to Figs. 2(a),(e), and (g). The delocalized regime with complex loops is demonstrated in blue, whereas the real spectrum in the localized regime is demonstrated in green color. The intermediate regime with partial real and complex eigenvalues is demonstrated in red. We have considered a lattice with 610 sites in all the three cases $V = 0.25$, 2.0 and 4.5.

## 4  Numerical results and discussions

In this section, we analyse the phase diagram of the DL transition in the presence of a strong interchain coupling between the two QHN chains A and B. The ratio of intrachain hopping strengths of the chains $A(B)$ is $t_L/t_R = 0.5$. In the upper panel of Fig. 2, we consider the case of symmetric interchain coupling. It is clear from Figs. 2(a)-(d) that the DL transition does not occur at $V_c = 2\max[J_R, J_L]$ (which is the critical value of DL transition in QHN chain). It is clearly visible that all the eigenstates are perfectly delocalized for $V \lesssim V_{c1}$ and localized for $V \gtrsim V_{c2}$. The value of $V_{c1}$ and $V_{c2}$ as determined in Eqs. (16) and (17) agrees excellently with all the numerical estimates. Furthermore, as is evident, the eigenstates between these two critical points are a mixture of both the delocalized and localized states, separated at a critical energy, termed as the mobility edge.

Next, we consider the case when the interchain coupling is asymmetric in nature in the lower panel of Fig. 2. It is clear that the localization behavior drastically changes upon considering a particular strength of asymmetricity, i.e, say, $u_L = 0.5$, $u_R = 1.0$ and $w_L = 0.5$, $w_R = 1.0$. Such a tendency of $V_{c1}$ approaching 0 is expected when $|t_L - u_2|$ and $|t_R - u_1|$ are both zero, as already explained. This particular case is of interest since the localized states appear even for a low value of the quasiperiodic potential, similar to the 1D original Anderson model, although in this case not all the states are localized.

To have a closer look into the nature of states in between $V_{c1}$ and $V_{c2}$, we check the fraction of localized ($\phi_l$) and delocalized states ($\phi_d$). We consider the states with $IPR \gtrsim 0.1$ as being absolutely localized, and below the limit the states are considered to be delocalized. We examine these different regions separately by plotting the fraction of localized and delocalized states as a function of the quasiperiodic potential corresponding to the parameters of Figs. 2. From Figs. 3(a-h), we can easily infer that there is a co-existence of localized and delocalized states for a wide regime in the quasiperiodic potential. Moreover, interestingly 50% of these states are delocalized while the remaining states are localized. This proportionate behaviour is consistent throughout the entire intermediate region. Furthermore, it is also important to see that in the case where $V_{c1} = 0$, exactly 50% localized states appear at even a tiny quasiperiodic potential, as previously discussed.

To infer the nature of the energy spectrum in the complex plane, in Figs. 4(a-c), we have plotted the eigenenergies in the three important regimes of the phase diagram as discussed in Fig. 2. We find that for any strength in the symmetric/asymmetric interchain coupling, in the

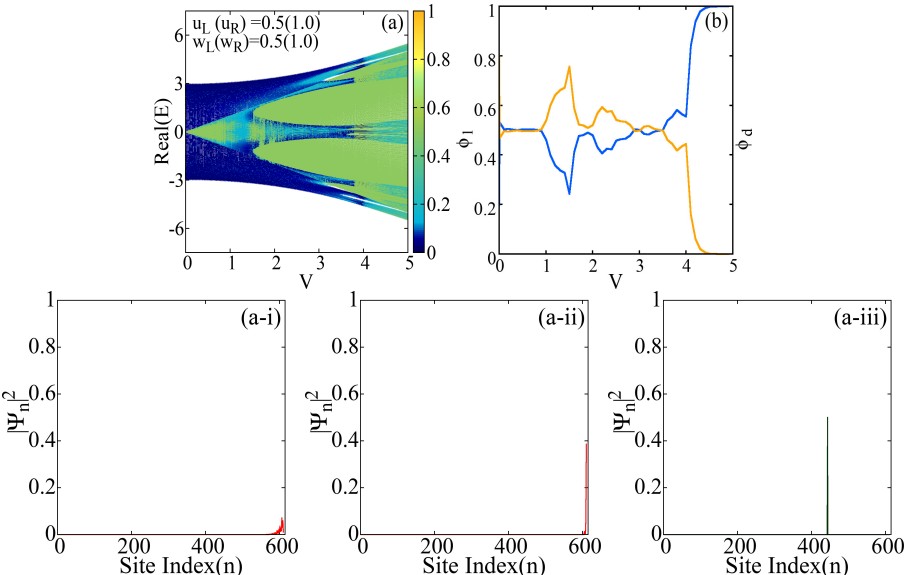

Figure 5: The localization behavior in a strongly coupled QHN Hamiltonian with $t_L = 0.5, t_R = 1.0$ for a lattice with $N = 610$ sites under OBC with interchain hopping between the two adjacent unit cells. (a) Projection of $IPR$ as a function of the real part of the eigen-energy and quasiperiodic potential $(V)$, where the DL transition is shown as a dark blue to green color transition. The other parameters of interchain coupling are: (a)$u_L = 0.5, u_R = 1.0, w_L = 0.5$ and $w_R = 1.0$. (b) The fraction of localized states $(\phi_l)$ in blue and delocalized states $(\phi_d)$ in yellow, corresponding to the parameters of Fig 2. Figs. a(i-iii) in the lower panel demonstrates the behavior of the wavefunction probabilities at different latice sites corresponding to the figure in the upper panel at $V = 1.5$. (a-i) skin modes (dark blue regime of $IPR$), (a-ii) skin modes in the light blue regime of $IPR$, and (a-iii) localized modes (in light blue regime of $IPR$).

delocalized regime the energy spectrum is completely complex and forms loops. On the other hand, when the states are entirely localized, the energy spectrum becomes completely real. This is in agreement to the previous observations demonstrated since early works ( [38, 39]). In the intermediate regime possessing mobility edges, we illustrate the presence of partially complex and real eigenenergies which arises due to the co-existence of delocalized and localized eigenstates. In addition, the energy spectrum under the OBC with similar features has been elucidated in Appendix A.

In Fig. 2(g), we find an interesting outcome of localization at a very minute value of the quasiperiodic potential $V$ under the PBC. However, it is well-known that the non-Hermitian systems with asymmetric hopping are drastically sensitive to the choice of boundary conditions, wherein the bulk states under PBC turn into skin states localized towards one of the boundaries under the OBC. Therefore, it becomes important to understand whether the same localization behavior is retained under the OBC in the coupled QHN system and to understand the nature of skin states. We plot the phase diagram for the interesting case corresponding to Fig. 2(g) under the OBC in Fig. 5(a), which exhibits qualitative distinctions in the delocalized/intermediate/localized regimes as compared to its PBC counterpart. It is clear that some of the localized states under PBC (in green) turn into states with IPR values lying in the light blue regime. For more clarity about the exact nature of the eigenstates, we plot the percentage of the delocalized and localized eigenstates in Fig 5(b). One can immediately infer that the proportion of delocalized and localized states does not remain same (i.e., at 50%) when the

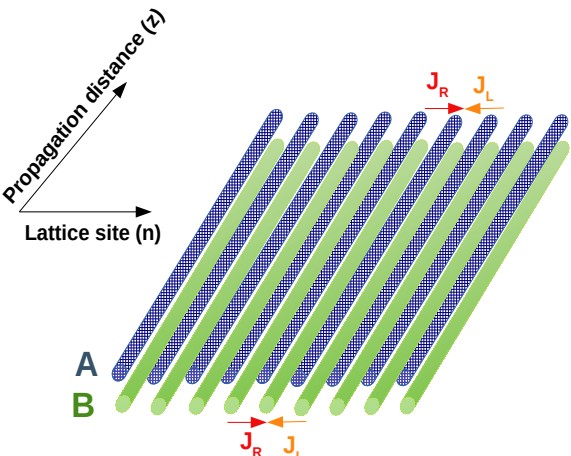

Figure 6: Schematic diagram of the bilayaer coupled waveguided array with asymmetric hopping. Atoms *A* and *B* which depict the waveguide channels in the optical set-up are depicted in blue and green respectively.

boundaries are open. It is clear, that the localized wavefunctions under PBC become delocalized(or skin) under the OBC since the fraction of delocalized states($\phi_d$) significantly increases. We plot the wave-function probabilities at different regimes of the phase diagram in Figs. 5(a-i)-(a-iii). In Fig. 5(a-i), we pick one of the eigenstates from the dark blue regime of the phase diagram, where we find the existence of a skin state under OBC as expected. Interestingly, we find that the light blue regime infact consists of both skin states (localized at right edge as demonstrated in Fig. 5(a-ii)) and localized states where the localization is not necessarily towards the right edge as shown in Fig. 5(a-iii). This is in stark contrast to the 1D HN systems in the absence of the interchain coupling, where the localized states under PBC remain localized even under the OBC. One can therefore infer that additional skin modes are formed from the localized states under OBC due to the coupling between such QHN chains, and hence the one-to-one correspondence between the delocalized(skin) states under PBC(OBC) breaks down in the presence of the coupling.

# 5 Possible experimental implementation in coupled waveguides

The equation of a coupled waveguide array at position *n* is written in the form,

$$-i\frac{d\psi_n}{dz} = t_L\psi_{n+1} + t_R\psi_{n-1} + V_n\psi_n, \tag{18}$$

where $t_L$ and $t_R$ tune the spacing in between the waveguides, and is non-Hermitian in the usual sense. Eq. 18 is an optical analogue of the Schrodinger equation where the time *t* is replaced by propagation distance along the parallel waveguides *z*, due to the mathematical equivalence between the two [43,44]. In this work, starting from the two coupled QHN chains, we get two coupled equations (Eq. 11 and Eq. 12 for the two atoms in the sublattice). We will therefore obtain two equations similar to Eq. 18 for A and B atoms coupled to each other, given as,

$$-i\frac{d\psi_{n,A}}{dz} = t_L\psi_{n+1,A} + t_R\psi_{n-1,A} + w_L\psi_{n+1,B} + u_R\psi_{n-1,B} + V_n\psi_{n,A}, \tag{19}$$

and

$$-i\frac{d\psi_{n,B}}{dz} = t_L\psi_{n+1,B} + t_R\psi_{n-1,B} + w_R\psi_{n-1,A} + u_L\psi_{n+1,A} + V_n\psi_{n,B}. \tag{20}$$

Intuitively speaking, since we have two atoms (A and B) in a unit cell, we have two layers of waveguided arrays (each of them being termed as a coupled waveguided array) as depicted in the schematic given in Fig. 6. Such a single coupled waveguided array has already been fabricated experimentally on a semiconducting AlGaAs substrate when $t_L = t_R$ [44]. It has been demonstrated that the array is composed of a core layer sandwiched between two cladding layers, where the upper cladding layer is etched quasiperiodically, where one can modulate the width of the waveguides quasiperiodically to realize the quasiperiodic onsite potential. The etching makes the core beneath it have a lower effective refraction index, resulting in a array of coupled 1D waveguides. One can now tune $t_L$ and $t_R$ (as required in our work) using a beam-splitter. Because of the two different atoms, we consider another coupled waveguide placed exactly below it, which could mimic the coupled QHN system as discussed in our main text. Since our work demonstrates the avenue to tune the strengths of $V_{c1}$ and $V_{c2}$ to engineer the localization transitions, such a coupled waveguided array can prove to be a boon to experimentalists working in such optical set-ups.

## 6  Conclusions

To summarize, this work scrutinizes the different localization attributes in non-Hermitian coupled quasiperiodic chains. The nature of DL transition at a threshold of the quasiperiodic potential ($V_c = 2$) for NH-AAH chains with parameters $t_L = 0.5$ and $t_R = 1.0$ is well-known. However, unlike the generic DL transition in NH-AAH chains, a strong coupling between the atoms of adjacent unit cells of the two HN chains possessing the same directionalities under PBC renders an intermediate region, wherein the eigenstates are a mixture of equal proportion of delocalized and localized states. Interestingly, for the counterpart with asymmetricity with specific hopping amplitudes, this intermediate region appears even in the presence of very tiny quasiperiodic potential, where the localized and delocalized states coexist. In this case as well, the proportion of localized and delocalized states remains identical. Moreover, under an OBC, we find a mixture of skin states and localized states in a regime of the localized portion in the PBC phase diagram. This is in contrary to the conventional HN systems where the localized states under OBC can either be skin modes or be completely localized and the usual PBC-OBC correspondence that leads the delocalized states to become skin states, keeping the localized states intact completely breaks down in the presence of the coupling. We believe that these rich phases due to the coupling in non-Hermitian systems can be utilised in experiments related to coupled waveguides.

## Acknowledgments

The authors are thankful to the High Performance Computing (HPC) facilities of the National Institute of Technology (Rourkela).

**Funding information**  The computation was carried out in the cluster procured from SERB (DST), India (Grant No. EMR/2015/001227). A.C. acknowledges CSIR-HRDG, India, for providing financial support via File No.- 09/983(0047)/2020-EMR-I.

## A Energy spectrum of the coupled QHN model under OBC

Since the non-Hermitian systems with asymmetric hopping amplitudes are sensitive to the boundary conditions, in Fig. 7, we plot the energy spectrum in the three regimes (*i.e.,* skin, localized and intermediate regimes) under OBC. The nature of the localized and intermediate states are identical to those under the PBC (Fig. 4). Interestingly, in the regime where the skin states are present (as illustrated in Fig. 5 of the main text), the energy spectrum remains complex. Such a formation of skin states with complex energies has also been observed in recent works [33, 45]. Moreover, in this regime, a few scattered energies inside the complex loops arise due to edge effects because of the finite size of the system considered in our work, which is frequently observed in such kind of non-Hermitian systems. In addition, the partial reality in energy spectrum as observed in the upper panel of Fig. 7(c) clearly illustrates the intermediate regime for the special strength of asymmetric interchain hopping as described in the main text.

## B The behavior of IPR with varying system sizes

To comprehend the system size dependence of the IPR with varying system sizes and the reason behind selection of $IPR = 0.1$ criterion for the separation of delocalized and localized eigenstates, we have plotted the IPR as a function of the normalized eigenstate index in the delocalized and localized regimes at a particular strength of the inter and intrachain coupling amplitudes in Figs. 8(a-b). It is evident that in the delocalized regime, the IPR decreases with an increase in the size of the system, whereas in the localized regime it remains constant and overlaps for all system sizes. In the intermediate regime, the delocalized states exhibit the system size dependence, whereas the localized states remain pinned at a higher value of IPR close to 1. Furthermore, we have verified this for all other interchain and intrachain coupling amplitudes and under the OBC. These results which do not convey any additional information have not been demonstrated here for brevity. Moreover, it is clear from Figs. 8(a-b) that the threshold $IPR = 0.1$ is appropriate for the clear distinguishability of the delocalized states from the localized ones at all system sizes and is therefore considered in our work to estimate the accurate percentage of delocalized and localized eigenstates in the intermediate regime.

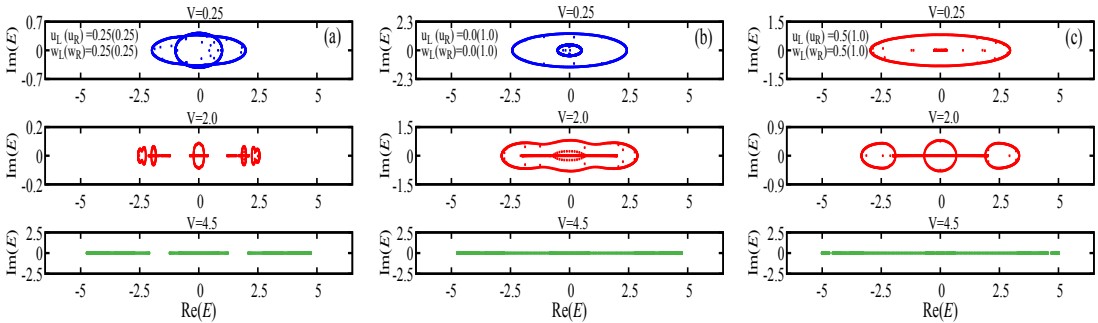

Figure 7: (a), (b), (c) The energy spectrum in the complex plane under the OBC corresponding to Figs. 4(a),(b), and (c). The rest of the parameters are same as in Fig. 4.

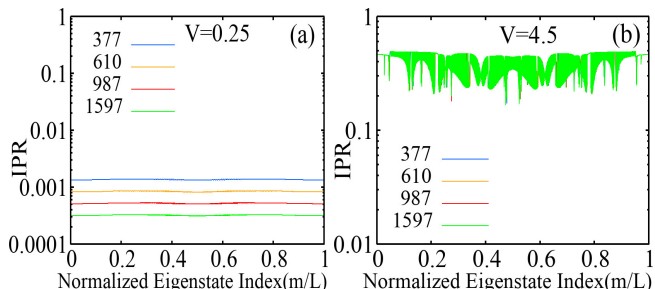

Figure 8: IPR as a function of the normalized eigenstate index (n/L) under PBC in the (a) delocalized ($V = 0.25$) and (b) localized($V = 4.5$) regimes with four different system sizes ($N = 377, 610, 987, 1597$). The other inter/intrachain coupling amplitudes are: $t_L = 0.5, t_R = 1.0,$ and $u_L = u_R = w_L = w_R = 0.25$.

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
