# Peer review of "Engineering unique localization transition with coupled Hatano-Nelson chains"

_SciPost Physics Core, doi:SciPost Phys. Core 8, 033 (2025)_

## Round 1 · Referee Report · Anonymous (Referee 1) · 2024-12-21

Report

The authors investigate the localization properties of coupled one-dimensional quasiperiodic Hatano-Nelson (QHN) chains with asymmetric hopping. They identify two critical points, $V_{c_1}$ and $V_{c_2}$ , which mark transitions between delocalized and localized states. For potential values $V<V_{c_1}$ the system exhibits complete delocalization, while for $ V>V_{c_2}$ all states become localized. Between these two points, mobility edges arise, separating delocalized and localized states. The authors further demonstrate that under specific asymmetric hopping conditions,$V_{c_1}$ can approach zero, resulting in localization even for infinitesimally weak potential amplitude. These findings are derived analytically by extending results from prior works and validated through numerical analysis. A potential experimental implementation using coupled waveguide arrays is also referred.
While these results contribute to the active field of non-Hermitian disordered physics, where numerous studies on variants of quasiperiodic models have been conducted, the manuscript suffers from clarity, language, and scientific presentation issues. At this point, I would not recommend publication in SciPost Physics unless the authors provide satisfactory responses or corrections to the issues outlined below.

1. The spectrum of the stacked Hatano-Nelson bilayer is not clearly explained. Beyond plotting the IPR as a function of the real part of the eigenvalue and increasing quasiperiodic potential amplitude, the manuscript should include comments on the structure of the spectrum itself, and whether and how it depends on the applied boundary conditions. The boundary conditions for Figures 2 and 3 are not specified in the main text (e.g., after Eq. 6). Additionally,$V_{n}$ (Eq. 7) is not defined.
2. Fig. 2 / Lines 130-131: In several subplots of Fig.2, for $V<V_{c_1}$ , certain energies are not represented by dark blue, indicating that the eigenstates are not “perfectly” delocalized as claimed in line 131. Also, how do the values of IPR in the delocalized/intermediate/localized phases change with increasing lattice size N?
3. The criterion IPR>0.1 is arbitrary and does not strictly justify labeling states as “delocalized” or “localized”. How does this choice affect the fraction of localized and delocalized eigenstates in the “localized”, “intermediate” and “delocalized” regimes of potential amplitude? The fraction of 0.5 in the intermediate phase would likely differ with a different choice of criterion. To what extent does this choice influence the universality of the results (e.g., the constancy of the fraction in the intermediate phase)?
4. The meaning of lines 157-167 is unclear due to the expression and complicated structure. It is difficult to understand what the authors are trying to convey and how these results differ from those obtained under PBC. Additionally, the purpose of Figures 4. a(i, ii, iii) is ambiguous. What specific eigenvalues of the spectrum do these figures correspond to?
5. The section “Possible Experimental Implementation in Coupled Waveguides” reformulates the problem within the context of coupled-mode theory as an example of a realistic experimental setup in optics. Equation (18) pertains to a single waveguide lattice, not the bilayer lattice model discussed previously. Furthermore, the optical analog of time is the propagation distance z along the waveguides (as shown in Fig. 5) and not the distance between two parallel waveguides, as stated in line 172.

Recommendation

Ask for major revision

  • validity: good
  • significance: good
  • originality: good
  • clarity: low
  • formatting: below threshold
  • grammar: below threshold

---

## Round 2 · Referee Report · Rudolf Roemer (Referee 2) · 2025-2-25

Strengths

This seems to be a solid manuscript on an interesting topic with a good and understandable exposition of the theoretical background, the expectations and the results obtained.

Weaknesses

There is nothing that feels new or exceptional: The HN model is previously well-known, the analytical and numerical methods are also well studied and their use for non-hermitian situations is established.

Report

In order to be publishable in SciPost Physics, the authors have chosen

  • Open a new pathway in an existing or a new research direction, with clear potential for multi-pronged follow-up work
  • Details a groundbreaking theoretical/experimental/computational discovery

I don't think either is right. The possible pathways that open with the non-hermiticity are similar to the multitude of works that have emerged recently using non-hermiticity. The authors have merely identified a model in which this has not yet been done before. Yes, this is interesting, but not very much so. I see no evidence for groundbreaking work, neither in model, methods nor results.

Requested changes

-

Recommendation

Accept in alternative Journal (see Report)

---

## Round 2 · Referee Report · Anonymous (Referee 1) · 2025-2-28

Strengths

The revised manuscript demonstrates improvements in clarity and overall presentation. The results of the paper contribute to the active field of non-Hermitian disordered physics.

Weaknesses

The manuscript still suffers from presentation issues. See for example Figures 4, 7 where the subfigures are misaligned with stretched tick labels in tiny fonts, and in Figure 8 the x-axis label is partly cut (see the letter “g”).

Report

The criteria claimed by the authors for submitting to SciPost Physics are not met, in my opinion, as the manuscript does not open a new pathway with clear potential for multi-pronged follow-up work, nor does it detail a groundbreaking discovery. However, I believe that the manuscript has interesting and original—though not groundbreaking—results in a very active field where numerous papers dealing with similar models have recently appeared, so in my opinion it does satisfy the criteria for publishing in SciPost Physics Core. I therefore suggest acceptance for publication in SciPost Physics Core.

Recommendation

Accept in alternative Journal (see Report)

---

## Round 2 · Author Response

We thank the anonymous referee for the insightful comments which helped us improve the overall quality of the manuscript. We have revised the article according to the suggestions of the referee and thoroughly revisited the points raised. The statements lacking clarity have now been updated.

Below, we mention our reply to the specific comments/queries of the referee:

  1. Comment/Query: The spectrum of the stacked Hatano-Nelson bilayer is not clearly explained. Beyond plotting the IPR as a function of the real part of the eigenvalue and increasing quasiperiodic potential amplitude, the manuscript should include comments on the structure of the spectrum itself, and whether and how it depends on the applied boundary conditions. The boundary conditions for Figures 2 and 3 are not specified in the main text (e.g., after Eq. 6). Additionally, $V_n$ (Eq. 7) is not defined.

Reply/Action: We have checked the nature of the spectrum of the coupled quasiperiodic Hatano-Nelson model under periodic boundary condition (PBC) and open boundary condition (OBC) for all the symmetric/asymmetric hopping parameters considered in Fig. 2 of our manuscript. Alike the previous observation in generic Hatano-Nelson systems with asymmetric hopping amplitudes, we find a complete complex-real transition in the energy spectrum that corresponds to the delocalized-localized phase transition under the PBC. Moreover, the spectrum consists of a mixture of complex and real eigenenergies in the intermediate regime due to the presence of both delocalized and localized eigenstates. This has been demonstrated for the certain parameters in Fig. 4 of the present version of the manuscript. Under the OBC, the nature of the spectrum remains almost identical, except a very few points present inside the complex spectrum of the delocalized regime (which under OBC becomes the regime with skin effect), anticipated due to the edge effects because of the finite size of the system considered in our work. The results under OBC have been presented in Appendix A. Such a presence of skin effect with complex energy spectrum has also been reported earlier, as referred to in the manuscript. Figs. 2 is estimated using the PBC as mentioned in the figure caption. Fig. 3 corresponds to the same parameters as in Fig. 2. We have now explicitly written the boundary condition (PBC) in the figure caption. The analytical expressions of $V_{c1}$ and $V_{c2}$ obtained using the transfer matrices are achieved under PBC. It has now been explicitly mentioned in the main text. $V_n$ in Eq. 7 has also been properly defined.

  1. Comment/Query: Fig. 2 / Lines 130-131: In several subplots of Fig.2, for $V < V_{c1}$, certain energies are not represented by dark blue, indicating that the eigenstates are not “perfectly” delocalized as claimed in line 131. Also, how do the values of IPR in the delocalized/intermediate/ localized phases change with increasing lattice size N?

Reply/Action: In Fig. 2, in the regime when $V < V_{c1}$ , the states are indeed delocalized and therefore appear as dark blue colored patches. The white color in the background which is mistaken for other color actually indicates the absence of any state with those particluar values of the real part of eigenenergy. Therefore, line numbers 130-131 in the previous version of the manuscript have not been changed. The behavior of the IPR in the delocalized/ localized phases with increasing lattice size N has been demonstrated in Fig. 8 of Appendix B. The IPR in the delocalized states decreases as N increases, whereas it remains constant over N in the localized regime as expected. In the intermediate regime, at some m/L values the IPR will therefore vary inversely with the lattice size N (delocalized states), and over other values of m/L it is expected to remain constant (localized states). This is expected and is also observed, but is not presented for brevity.

  1. Comment/Query: The criterion IP R > 0.1 is arbitrary and does not strictly justify labeling states as “delocalized” or “localized”. How does this choice affect the fraction of localized and delocalized eigenstates in the “localized”, “intermediate” and “delocalized” regimes of potential amplitude? The fraction of 0.5 in the intermediate phase would likely differ with a different choice of criterion. To what extent does this choice influence the universality of the results (e.g., the constancy of the fraction in the intermediate phase)?

Reply/Action: We have illustrated the system size dependence of the IPR in the delocalized and localized regimes of the coupled QHN system in Fig. 8 of the revised version of the manuscript, with its detailed discussion in Appendix B. It is clear that the delocalized eigenstates have IP R < 0.1, which keeps on decreasing with N . However, the IP R in the localized regime lies much away from 0.1 and is in fact closer to 1, which is also well known from the existing literature. We find some fluctuations at different values of m/L in the localized regime, but at all these states the IP R is much greater than 0.1. Therefore, for the system size considered in this work (N = 610), the threshold of 0.1 serves best to identify the nature of eigenstates. The fraction of 50% delocalized and localized states in the intermediate phase will also not be affected since the IPR values of both delocalized and localized eigenstates lie far away, retaining the universal nature in our results regarding the constancy of fraction of delocalized and localized eigenstates in the intermediate regime. One will therefore always obtain 0.5 as long as the threshold is selected in between the highest value in IPR of the delocalized states and the lowest value in IPR of the localized eigenstates.

  1. Comment/Query: The meaning of lines 157-167 is unclear due to the expression and complicated structure. It is difficult to understand what the authors are trying to convey and how these results differ from those obtained under PBC. Additionally, the purpose of Figures 4. a(i, ii, iii) is ambiguous. What specific eigenvalues of the spectrum do these figures correspond to?

Reply/Action: Since the localization properties under OBC in such non-Hermitian systems differ drastically from those under PBC, we have tried to construct a phase diagram in Fig. 4 (now Fig. 5) similar to Fig. 2. We find that in the most interesting case where the localization appears at a minute strength of the quasiperiodic potential (corresponding to Fig. 2(g)), few localized states actually become skin states (localized towards the right boundary) under the OBC. Figs. 5a(i-iii) demonstrate the presence or absence of skin states. In Fig. 4 a(i), we demonstrate the usual skin modes(which were delocalized under PBC). Figs. 5 a(ii) and (iii) belong to the states which were localized under PBC, but as one can see that under the OBC few states become skin states as shown in Fig. 5 a(ii), while the remaining are localized (Fig. 5 a(iii)). We have therefore demonstrated the non-trivial conversion of a few localized states under the PBC into skin states under the OBC, which is absent in the decoupled Hatano-Nelson chains. This has been clearly explained in details in the revised version of the manuscript. The eigenstates for which the probability distribution are infact picked up from the IPR values which lie in the dark blue, light blue and green regimes of the phase diagram under OBC respectively.

  1. Comment/Query: The section “Possible Experimental Implementation in Coupled Waveguides” reformulates the problem within the context of coupled-mode theory as an example of a realistic experimental setup in optics. Equation (18) pertains to a single waveguide lattice, not the bilayer lattice model discussed previously. Furthermore, the optical analog of time is the propagation distance z along the waveguides (as shown in Fig. 5) and not the distance between two parallel waveguides, as stated in line 172.

Reply/Action: We agree to the comment made that Eq. (18) refers to a single waveguide lattice and not the bilayer lattice. Eq. (18) is a general Schrodinger equation where time is replaced by the distance along the propagation in the waveguide. The idea in the experimental realization is to form a bilayer waveguided array with two coupled equations obtained in Eqs. (11) and (12), similar to Eq. 18. We have therefore added equations 19 and 20 for the bilayer lattice. We also accept our mistake in writing the propagation distance along the waveguide (z) to be the distance between the two waveguides. The discussion has therefore been slightly reformulated with the correct statement regarding ‘z’.

---

## Round 2 · List of Changes

1. Line numbers 58-62, 111-112, 159-168, in the present version of the manuscript. Appendix A and Figs. 4, 7 have also been added.
  2. No addition/modification in Fig. 2 and lines 130-131 are required. Appendix B now demonstrates the behavior of IPR in the delocalized and localized regimes with an appended figure therein (Fig. 8).
  3. Appendix B and Fig. 8 have now been added. 4.The discussion in line nos. 157-167 (in the earlier version) have now been updated for clarity. It has now been discussed in line nos. 169-194 in the present manuscript.
  4. Line nos. 196-217 have been reformulated.
  5. Additionally, we have also checked the manuscript to minimize the grammatical errors and enhance clarity.

---

## Round 3 · Author Response

List of changes
1. Figs. 4 and 7 have now been updated, where the sub-figures are now properly aligned. The font size has been increased.
2. The x label of Fig. 8 has been updated.
3. The references have been properly formatted.

---

## Round 3 · List of Changes

1. Figs. 4 and 7 have now been updated, where the sub-figures are now properly aligned. The font size has been increased.
2. The x label of Fig. 8 has been updated.
3. The references have been properly formatted.

---

## Editorial Decision

published